# Wound Repair and Ca^2+^ Signalling Interplay: The Role of Ca^2+^ Channels in Skin

**DOI:** 10.3390/cells13060491

**Published:** 2024-03-11

**Authors:** Gregorio Bonsignore, Simona Martinotti, Elia Ranzato

**Affiliations:** 1Dipartimento di Scienze e Innovazione Tecnologica (DiSIT), University of Piemonte Orientale, 15121 Alessandria, Italy; gregorio.bonsignore@uniupo.it (G.B.); simona.martinotti@uniupo.it (S.M.); 2SSD Laboratori di Ricerca—DAIRI, Azienda Ospedaliero-Universitaria SS. Antonio e Biagio e Cesare Arrigo, 15121 Alessandria, Italy

**Keywords:** Ca^2+^, Ca^2+^ channels, Ca^2+^ signalling, wound repair

## Abstract

The process of wound healing is intricate and tightly controlled, involving a number of different cellular and molecular processes. Numerous cellular functions, especially those related to wound healing, depend critically on calcium ions (Ca^2+^). Ca^2+^ channels are proteins involved in signal transduction and communication inside cells that allow calcium ions to pass through cell membranes. Key Ca^2+^ channel types involved in wound repair are described in this review.

## 1. Introduction

Wound healing is a complex and dynamic process that typically occurs in distinct phases [1]. 

In order to reduce blood loss, the first reaction to an injury is vasoconstriction and the creation of a blood clot. The initial clot is formed primarily by platelets, which also initiate the coagulation cascade. Next, debris, bacteria, and injured tissue are eliminated from the wound site by inflammatory cells that invade it, mostly neutrophils. Macrophages are then recruited to further clean the wound and release growth factors that initiate the tissue repair process [2].

The provisional matrix is formed by collagen, which is produced by fibroblasts that migrate to the wound site. The process of angiogenesis, or the growth of new blood vessels, supplies oxygen and nutrients to aid in tissue healing. As they multiply and spread out, epithelial cells cover the surface of the wound. Fibroblasts continue to deposit collagen and eliminate extra tissue, which results in the remodelling of the collagen matrix. Scar tissue grows and the strength of the wound rises [3]. 

## 2. The Role of Calcium Signalling during Wound Repair

In several phases of wound healing, calcium (Ca^2+^) signalling plays a crucial role in coordinating the cellular reactions necessary for tissue restoration. Processes including cell migration, proliferation, and extracellular matrix remodelling are coordinated by the dynamic fluctuations in intracellular calcium levels [4]. 

When a wound occurs, there is often a fast increase in intracellular calcium levels in the cells surrounding the injury site [5]. This rapid influx of Ca^2+^ can be triggered by mechanical forces, changes in membrane potential, or the activation of cell surface receptors in response to injury. Then, Ca^2+^ signals play a central role in guiding the migration of various cell types to the wound site, including fibroblasts, immune cells, and epithelial cells [6,7,8] (see Figure 1). 

Furthermore, the cytoskeletal dynamics required for cell migration and movement toward the injured location are regulated by calcium-dependent signalling pathways. Immune cells are drawn to the wound site during the inflammatory phase of wound healing in order to remove debris and stop infection. Immune cells are activated in response to calcium signalling, which aids in guiding their migration to the site of injury. Ca^2+^ signals also regulate the process of phagocytosis, which is crucial for the removal of dead cells, debris, and pathogens from the wound. Last but not least, intracellular calcium levels influence the cytoskeletal changes necessary for engulfing and digesting foreign particles [9]. Additionally, during the proliferative phase of wound healing, Ca^2+^ signalling stimulates cell multiplication. The regrowth of tissue is really made possible by elevated Ca^2+^ levels activating signalling pathways that promote cell division [10,11]. Ca^2+^ signals are also necessary for activating myofibroblasts, leading to the contraction necessary for wound closure [12]. Some calcium-dependent enzymes, such as matrix metalloproteinases (MMPs), are crucial for remodelling the extracellular matrix during the later stages of wound repair. So, Ca^2+^ signals regulate the activity of MMPs, influencing the breakdown of damaged matrix components and the deposition of a new matrix, allowing for correct wound and tissue repair [13]. The coordinated migration and division of epithelial cells at the margins of wounds during re-epithelialization are facilitated by calcium-dependent mechanisms [7].

In conclusion, the process of calcium signalling during wound healing is complex and dynamic. It plays a role in the timely repair of damaged tissue by participating in the following processes: cell migration, inflammation, proliferation, contraction, and extracellular matrix remodelling.

## 3. Calcium Channels Involved in Wound Repair

Ca^2+^ channels are essential components in wound repair, playing a crucial role in mediating the influx of calcium ions into cells. Various Ca^2+^ channels regulate the dynamic fluctuations in intracellular calcium levels, which aid in the cellular processes that are coordinated during the various stages of wound healing.

Here we describe key types of Ca^2+^ channels involved in wound repair (see also Figure 2).

### 3.1. Voltage-Gated Calcium (VGCC) Channels

Voltage-gated calcium channels (VGCCs) are trans-membrane proteins. A membrane potential depolarization can activate them [14] causing the conversion of membrane electrical signals to intracellular Ca^2+^ transients [15]; this condition leads to the activation of several physiological events. 

VGCCs are composed of different subunits called α1, β, γ, and α2δ [16]. The α1 subunits possess four homologous components with six trans-membrane sections [17]; these include trans-membrane segments S1–S4 which form the voltage-sensing unit, whereas S5 and S6 comprise pore 1 [18]. Cell membranes depolarize, when stimulated, and as a result, action potentials happen. The changes in potential are perceived by the voltage-sensing module, which mediates Ca^2+^ influx [19]. Ca^2+^ has an important function as a universal second messenger; as a consequence, it can stimulate many physiological phenomena, including cell death, gene expression, hormone secretion, muscle contraction, and synaptic transmission [15,19].

VGCCs have 10 subtypes, and they can be combined into families: Cav1, Cav2, and Cav3. The first subfamily is composed of L-type calcium channels. They are stimulated at high voltage to perform long-lasting and large ion currents [20]. The second subfamily, Cav2, is divided into R-type, N-type and P/Q-type based on their functions [21,22]. Conversely, the final subfamily, Cav3 channels, is T-type Ca^2+^ channels that conduct transient currents when activated at low voltages [21,23]. It was discovered that VGCCs are located on epidermal keratinocytes. The Ca^2+^ ions’ influx into epidermal keratinocytes defers the barrier rescue (as demonstrated by Denda et al. [24]); on the contrary, nifedipine administration, R-(+)-BAY K8644 or verapamil, VGCC inhibitors, hastens the healing. When the skin barrier is damaged, its electric potential changes; this condition is able to generate negative electric potentials to help with the repair of the skin barrier [25]. According to these findings, VGCCs are linked to skin homeostasis and have a detrimental effect on the restoration of the skin barrier [26].

### 3.2. Transient Receptor Potential (TRP) Channels

Ion channels known as transient receptor potential (TRP) channels are a broad family that is involved in perceiving and reacting to a range of environmental stimuli, such as pressure, temperature, and chemical signals. These channels play a significant role in wound healing by contributing to processes such as inflammation, cell migration, and pain perception.

At first, TRP genes were described in *Drosophila melanogaster*. A study showed that in the sight-impaired mutant fly, there is a transitory response to continuous light [26] rather than a prolonged one [27]. However, the trp gene structure and the TRP protein localization were recognized later [28]. 

Currently, in humans, research has identified around thirty TRP-related genes and twenty-seven different TRP channels [29]. TRP channels play an important role in communication between cells; in addition, their activities are dependent on cell type. For example, in neural cells, the activation of TRP channels could induce depolarization and the induction of electric potential. Moreover, TRP channels control intracellular calcium concentrations in non-excitable cells. This mechanism is associated to growth and the differentiation of keratinocytes, thus influencing the skin barrier [30]. Originally, TRP channels were defined as “polymodal cellular sensors” [31,32,33]; now, they are recognized to be “promiscuous pleiotropic molecules” activated by several stimuli [34,35,36].

TRP channels are divided into various subtypes based on their structure, i.e., TRPN (Drosophila NompC), TRPA (ankyrin), TRPC (canonical), TRPM (melastatin), TRPML (mucolipin), TRPP (polycystin), and TRPV (vanilloid) [28,37,38]. Recently, Li et al. identified, in yeast, another TRP family and called it TRPY [29].

Based on their similarity to Drosophila TRP, the channels are classified into two groups: group 1, which includes TRPA, TRPM, TRRPV, and TRPC and shows the greatest parallelism with Drosophila trp genes; and group 2, which consists of TRPM and TRPP, TRP receptors that are more dissimilar to Drosophila TRP receptors [29]. TRP channels are implicated in the activation of immune cells during the inflammatory phase of wound healing. Some TRP channels, such as TRPV1, TRPV3, and TRPA1, are expressed in immune cells and can be activated by inflammatory mediators. The activation of TRP channels in immune cells can lead to the release of pro-inflammatory cytokines and the modulation of immune responses [39]. TRP channels are expressed in sensory neurons, and their activation can contribute to pain perception and the transmission of sensory signals associated with tissue injury. Other TRP channels, including TRPV4, have been implicated in the regulation of blood vessel function. TRPV4 activation, for instance, can lead to the release of vasodilatory factors, influencing blood flow to the wound site and supporting the delivery of nutrients and immune cells [40].

Based on available data, it appears that many TRP channels [41] are connected to cutaneous conditions like psoriasis, acne vulgaris, atopic dermatitis (AD), cutaneous malignancies, dermatitis of various kinds, and hair development abnormalities [42].

#### 3.2.1. TRPC

The TRPC group of receptors is composed of seven members (indicated as TRPC1 to 7), and they are strictly correlated to TRP channels described in *Drosophila*. These receptors, which are non-specific cation channels found in both nervous and non-nervous cells, play a crucial role in controlling intracellular Ca^2+^ entry in response to both normal and pathological stimuli [43]. An increase in Ca^2+^ entry leads to membrane depolarization and a rise in the cytosolic Ca^2+^ concentration ([Ca^2+^]_c_), both of which have detrimental consequences on cellular functions [44]. Research has highlighted the TRPC’s role in myocardial injuries and in diabetic kidney diseases [45,46]. TRPC inhibition shields the podocytes and cardiac cells’ activity; as a result, TRPC could be an auspicious target. In addition, TRPCs are similarly localized in the epidermal cells [47], and some authors have shown an involvement in Darier’s disease by the modulation of keratinocyte differentiation and proliferation [48].

During foetal development, TRPC1 expression is higher in the liver and kidney than in the brain; however, following birth, it is primarily found in the ovary, heart, testes, and some areas of the brain [49]. 

In humans, TRPC2 is considered a pseudogene [28]. TRPC1 contributes to the differentiation of keratinocytes through the regulation of the Ca^2+^ influx [50]. TRPC1 on the endothelial cell membrane is able to interact with both vascular endothelial growth factor (VEGF)/VEGF receptor 2, controlling plasma membrane permeability and Ca^2+^ influx, and with soluble α-Klotho (e.g., vascular-protective and antioxidant molecule), thus playing a significant role in the maintenance of endothelial cell integrity [51]. 

TRPC6 expression, instead, is facilitated by non-canonical TGF-β signalling through p38 and Serum Response Factor (SRF). TRPC6 triggers the calcineurin (a calcium-sensitive phosphatase), which causes trans-differentiation induction in myofibroblasts [52]. The granulosum and spinosum stratum keratinocytes are the primary epidermal cell types that contain TRPC6. However, TRPC6 is not expressed in basal layer keratinocytes, because this channel is able to modulate Ca^2+^-dependent differentiation [53]. In addition, hyperforin, contained in Saint John’s wort, can induce TRPC6 to improve ATP-Ca^2+^ signalling and Ca^2+^ entry in skin cells, thus participating in skin repair [54]. In the same way, some data suggest that in gingival keratinocytes, TRPC4 is present in the cytoplasm and in the plasma membrane, regulating the calcium-sensing receptor (CaSR)-induced increase in intracellular [Ca^2+^] [55].

Additionally, certain TRPC group members can interact with one another and even perform sophisticated functions; for example, TRPC3 can take action as a STIM1 (stromal interaction molecule)-dependent store-operated Ca^2+^ channel (SOCCs) only, interacting with TRPC1, while, in the presence of TRPC4, TRPC6 acts only as a STIM1-dependent channel [56]. Through changes in the Ca^2+^ level, STIM1 affects TRPC4 and TRPC1, which in turn control store-operated Ca^2+^ entry and impact the maturation of epidermal cells and the establishment of the epidermis barrier [57].

#### 3.2.2. TRPV

TRPVs are responsive to several signals coming from tissue damage and their trigger is often recognized as pain. According to the literature, thermo-TRP channels such as TRPV1-4 are triggered by heat. The TRPV is another subgroup of TRP channels, and it is composed of six members (TRPV1-6) [29]. On the cells of basal and supra-basal healthy human skin, TRPV1-4 immunoreactivity has been recognized, and these groups have been suggested to be thermo-sensory receptors [58,59]. TRPV5 and TRPV6 have been recognized as calcium ion channels present in epithelial cells; on the contrary, TRPV1-4, considered nociceptors, are able to sense the damaging signals [60]. In addition to sensing protons (low pH), toxic heat, and capsaicin, TRPV1 may also react to UV stimulation [61]. Hair shaft elongation and keratinocyte proliferation are inhibited by TRPV1 activation. This condition conducts early hair follicle regression and consequent cellular death by apoptosis [62]. The AEA (anandamide) application promotes cell death by Ca^2+^ entry through TRPV1 and blocks epidermal proliferation by the increase in intracellular Ca^2+^ levels [63]. In developed human primary keratinocytes, TRPV1 inhibition prevents PAR-2 (proteinase-activated receptor-2) from triggering the SLIGKV peptide, which causes the depletion of the Ca^2+^ storage and the release of inflammatory mediators [64]. These works show that the over-activation of TRPV1 inhibits keratinocytes’ proliferation during inflammation.

TRPV is regulated by temperature, by ligands, such as probenecid and cannabinoids, and by lipids. TRPV2 and TRPV1 have in common a high sequence identity (>50%), but TRPV2 displays a higher sensitivity for activation and the temperature threshold than TRPV1 [65]. However, different from TRPV1, there is little information regarding TRPV2 expression in human epithelial cells. Furthermore, no information on epithelial cells is available on the physiological role of TRPV2 [59]. Nevertheless, in vitro studies (on rat models) about wound healing have shown that compounds targeting TRPV2 channels could improve excessive wound contraction through the differentiation of dermal fibroblasts and the blocking of the release of TGF-β1 [66]. 

TRPV3 is mostly found in skin keratinocytes, where it senses pain and warmth. It is sensitive to warm temperatures (below 33 °C) [67,68]. TRPV3 activation enhances prostaglandin E2, nerve growth factor, thymic stromal lymphopoietin (TSLP), and interleukin (IL)-33 production in human keratinocytes and increases scratching behaviour in mice [69]. The triggering of TRPV3 can also encourage an important pro-inflammatory response by the means of the NF-κB pathway [70]. Some works have also described that in the tissues of pruritic burn scars, TRPV3 and TSLP expression have improved [71]; moreover, TRPV3 stimulates the differentiation of myofibroblast, collagen secretion, and the expression of TSLP via the TRPV3-Smad2/3 signalling pathway [72]. These results propose a link between pruritus diseases and TRPV3. For this reason, TRPV3 inhibition could be of great interest for therapy. Also, TRPV4 can also be triggered by temperatures under 33 °C [73,74], and it was originally portrayed to be a mechano- or osmo-sensor [75]. 

TRPV4 is largely expressed in the whole body [76]. TRPV4-deficient mice show an impaired epidermal barrier, non-physiological actin rearrangements, insufficient stratification, and leaky cell–cell junctions [77]. Ammar et al. discovered that the migration and the proliferation of wild-type oesophageal keratinocytes were slower than in those with TRPV4 knockout [78]. Moreover, TRPV4 has been emphasized [79] as being required for the TGF-β1-induced differentiation of cardiac fibroblasts into myofibroblasts. The siRNA knockdown of a TRPV4-specific antagonist, called AB159908, dramatically blocked TGFβ1-induced differentiation. TRPV5 and TRPV6 are both present at the apical portion of the cell membranes of Ca^2+^-transporting epithelia, and they are calcium-selective channels, helping as access receptors in trans-epithelial Ca^2+^ passage [80]. 

TRPV6, which participates in the formation of the skin barrier, plays an important function, under high extracellular Ca^2+^ levels, in the terminal differentiation mechanism [81]. Additionally, TRPV6-deficient keratinocytes display a decrease in the ability to flatten the strong connections between neighbouring cells [82].

#### 3.2.3. TRPA

Only one member of this family, TRPA1, is found in sensory neurons as well as non-neuronal cells (including epithelium and hair cells), and it can be activated by harmful external stimuli and low temperatures [83]. Furthermore, TRPA1 regulates the biological function of epidermal cells, like the TRPV subfamily members [42]. For example, the TRPA1 topical application agonists showed increased barrier recovery [84]. A recent work also demonstrated that the inhibition of TRPA1 can reduce skin oedema and pro-inflammatory cytokine levels [85]. 

Furthermore, activated TRPA1 is correlated with irritant contact dermatitis symptoms, such as itching, pain, and neurogenic inflammation [86]. All this information suggests that TRPA1 plays a central role in inflammation and skin injury.

#### 3.2.4. TRPM

The TRPM subfamily is composed of eight variable members, intrinsic membrane proteins, i.e., TRPM1-8 [87]. Except for TRPM4 and 5, TRPMs are Ca^2+^-permeable cation channels [88,89]. Human epidermal melanocytes express TRPM1, which has been demonstrated to be essential for the pigmentation process [90,91]. Similar to how TRPV1 is an extremely effective heat detector, TRPM8 is more susceptible to cold in the environment [30]. Contrarily, as TRPM8 is not activated by dibutyl phthalate, it is unlikely to contribute to phthalate-induced cutaneous hypersensitivity [92]. Until today, TRPM8 did not show a link to skin homeostasis or dermatitis [93]. However, the activation of it can prevent the production of CGRP in colon tissue and stop chemically induced irritation [94]. Additionally, it causes a rise in pro-inflammatory cytokines in the blood of normotensive rats [95].

### 3.3. Inositol 1,4,5-Trisphosphate Receptor and Ryanodine Receptor

Ca^2+^ release from intracellular storage, mostly ER/SR, is mediated by intracellular ligand-gated Ca^2+^ release channels. Two strictly related groups of intracellular Ca^2+^ release channels have been identified: the ryanodine receptor (RyR) and the inositol 1,4,5-trisphosphate receptor (IP_3_R). Although they are found in all cell types, the Purkinje cells of the cerebellum have the highest densities of IP_3_R channels, whereas the RyR is the main Ca^2+^ release channel in striated muscle.

There are various functional matches between RyR and IP_3_R channels [96] thanks to similar physiology and structural homology. In cells, the concentration of Ca^2+^ in the cytoplasm is about 100 nM, lower than the level of Ca^2+^ in the ER Ca^2+^ store and extracellular Ca^2+^ concentration. Cytoplasmic Ca^2+^ concentration may be momentarily increased by the trigger of RyR and IP_3_R. The activity of IP_3_R channels is regulated by a coupled interaction between the binding of its primary ligands, IP_3_ and Ca^2+^. Furthermore, in response to a variety of extracellular cues, including hormones, growth factors, light, neurotrophins, neurotransmitters, and odorants, IP_3_ is a second messenger produced by phosphoinositide turnover [97]. TRPC1 cooperates with IP_3_R and PLCgamma1, inducing the SOC activation in human skin cells as indicated by Tu et al. [47]. In addition, Schmitt et al. [98] recently demonstrated that phospholipase C (PLC) inhibition blocks IP_3_R Ca^2+^ release. When IP_3_ binds to IP_3_R, it releases Ca^2+^ from intracellular storage and raises the concentration of free Ca^2+^ in the cytoplasm, which stimulates a variety of cellular processes, from growth to death and from secretion to contraction [99].

Additionally, IP_3_R controls Ca^2+^ in a biphasic manner, meaning that it is activated at low concentrations (up to 0.3 μM) and suppressed at higher values (0.5–1 μM) [100]. However, IP_3_R function is also controlled by accessory Ca^2+^-independent proteins, redox potential, ATP, and Mg^2^ [101]. In the nervous system and muscle, RyRs have a prominent part in intracellular calcium level regulation. RyR1 is hardly found in epidermal differentiated layers, according to Sumiko et al. [84]. RyR2 is similarly found in differentiated sheets, specifically in the border layer between the stratum corneum and the granular layer. 

Moreover, RyR3 is found on skin cells; however, differentiated layers express it more strongly. Another study highlighted that the RyRs’ inhibition can promote in vivo wound repair, while in a keratinocyte cell line (i.e., HaCaT cells), wound closure is enhanced by an RyR antagonist, dantrolene [102]. These studies propose that in the epidermis, RyRs are linked with both epidermal barrier homeostasis and the differentiation of epithelial cells. In addition, some therapeutic effect may be obtained by the application of RyR antagonists.

### 3.4. Store-Operated Ca^2+^ Channels

The release of Ca^2+^ from ER could be activated by IP_3_ (inositol 3-trisphosphate), a crucial second messenger, producing ER Ca^2+^ release to the cytosol, as the intracellular Ca^2+^ main store [103]. According to a hypothesis on capacitive Ca^2+^ entry, the plasma membrane’s Ca^2+^ channels are activated when Ca^2+^ reserves are emptied, helping to replenish the stores. This supposition was later called SOCE (store-operated Ca^2+^ entry) [104]. Different works also recognized that SOCE not only allows Ca^2+^ to refill deposits but can produce constant Ca^2+^ signals itself that induce such crucial activities as cellular metabolism, exocytosis, and gene expression [103,105]. On the other hand, some proteins are identified as Ca^2+^ sensors for SOCE; they are called stromal interaction molecules (STIMs, in particular STIM1 and STIM2). When Ca^2+^ store reduction is perceived, STIMs will move to the cell membrane and turn on ORAI Ca^2+^ channels, the store-operated Ca^2+^ channels [104,106,107]. Furthermore, SOCE can be activated by STIM-activating enhancers. For example, STIMATE, which is found in the ER membrane, can migrate to ER–PM junctions and assist SOCE there [108]. SOCE plays a role in several physiological events like cell metabolism, gene expression, and cancer development. 

ORAI1 was revealed to be largely located in the lower epidermal sheet where it works in regulating epithelial cells’ polarized motility and proliferation [109]. It was found that either STIM1 or ORAI1 knockdown can intensely impair SOCE; this condition causes early keratinocyte differentiation, the impaired expression of keratin1, and the blocking of HaCaT cells’ growth in low Ca^2+^ [110]. Moreover, reduced SOCE in ORAI1-, ORAI2-, and ORAI1/2-deficient immune cells, for instance neutrophils, affects numerous cell functions, such as leukotriene expression, degranulation, phagocytosis, and reactive oxygen species (ROS) generation [111]. 

In the same way, ORAI1 and ORAI2 can form a heteromorphic channel complex, where ORAI2 reduces ORAI1 function and mitigates SOCE, while ORAI2 optimizes SOCE magnitude to inflect immune reactions [112]. 

These findings suggest a potential relationship between ORAI proteins and the immunological and barrier functions of the skin. ORAI demonstrates the capacity to interfere with TRP channels. It was discovered that TRPV1 and ORAI1 connect and translocate closely to one another at the cell membrane. Rapid calcium-dependent ORAI1 inactivation is triggered by Ca^2+^ entering the cell through TRPV1 channels, and this controls cell mobility and wound healing [113].

### 3.5. Mechanosensitive Ion Channels

Various ion channels are sensitive to mechanical forces, and their activation can lead to an influx of calcium ions. Mechanosensitive channels play a role in transducing mechanical signals associated with tissue injury into biochemical responses, contributing to the early phases of wound repair [114].

Mechanosensitive ion channels are a class of membrane proteins that play a crucial role in sensing mechanical stimuli and converting them into electrical signals. These channels are found in various cell types and organisms, from bacteria to mammals, and they are essential for numerous physiological processes [115]. Mechanosensitive ion channels are typically integral membrane proteins that form pores across the cell membrane [116]. Changes in membrane tension or deformation can directly influence the conformation of these channels, leading to the opening or closing of the ion-conducting pore. The opening of mechanosensitive channels allows for the passage of ions (e.g., sodium, potassium, calcium) across the membrane, leading to changes in membrane potential and cellular responses. Mechanosensitive ion channels are diverse and can be classified into different families based on their structure and properties.

Piezo channels are a family of mechanosensitive ion channels that have been identified as key players in transducing mechanical forces into cellular signals. One important aspect of Piezo channels is their role in calcium entry. There are two known members of the Piezo family: Piezo1 and Piezo2. These channels are large trans-membrane proteins with a unique structure, forming a trimeric complex that functions as a mechanically gated ion channel [117]. Piezo channels are directly stimulated by mechanical stimuli, such as shear stress or membrane tension. The mechanical force applied to the cell membrane induces conformational changes in Piezo channels, leading to the opening of a non-selective cationic pore, allowing for the influx of ions. Piezo channels are permeable to calcium ions, and their activation results in a rapid and substantial influx of calcium into the cell [118]. During wound healing, cells need to migrate into the wounded area to replace damaged tissue and proliferate to fill the gap. Mechanosensitive ion channels, by sensing mechanical cues, can influence cell migration. Changes in mechanical forces associated with the wound may activate these channels, triggering signalling pathways that guide the movement of cells towards the wound site. Studies have shown that Piezo1 channels are involved in cell migration, which is crucial for closing wounds [117]. Additionally, they may play a role in the regulation of vascular development and blood flow, impacting the delivery of nutrients and oxygen to the wound site [119].

Mechanosensitive channels may contribute to the cellular responses that mediate tissue remodelling. The channels can be involved in transducing mechanical signals that guide the deposition and organization of extracellular matrix components [120].

## 4. Conclusions

Wound repair is often a burden to the patient, usually restraining well-being. Consequently, real wound repair, without scars, deserves to be an important aim in dermatology [121]. Calcium ions are recognized to play a pivotal role in cellular communication. Although efforts have been made to completely understand the role of Ca^2+^ in wound healing, we have limited information on how, in the wound bed, Ca^2+^ is controlled through various growth factors, vitamin D or hormones [122]. Similarly, research on the effects of Ca^2+^ released from platelets, scaffolds, or dressings on cellular activity, wound healing, and tissue remodelling is lacking, despite experimental models suggesting the critical importance of Ca^2+^ management at the wound site [123]. Some data show the potential of adding Ca^2+^ into functional dressings or incorporating the ion as a biomolecule in scaffolds to hasten wound healing.

In summary, certain studies have demonstrated the possibility of utilizing and regulating Ca^2+^ release from scaffolds in order to hasten the healing process of wounds [124]. However, more preclinical and clinical evidence are needed to support the results of the research on the ability to modify Ca^2+^ signalling in the wound area in order to improve tissue restoration and healing. 

## Figures and Tables

**Figure 1 cells-13-00491-f001:**
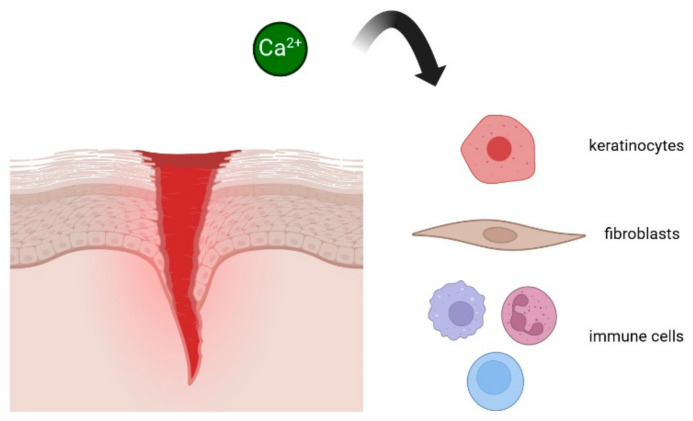
Ca^2+^ signals play a central role in guiding the migration of cell types to the wound site. See text for more details. Created with BioRender.com.

**Figure 2 cells-13-00491-f002:**
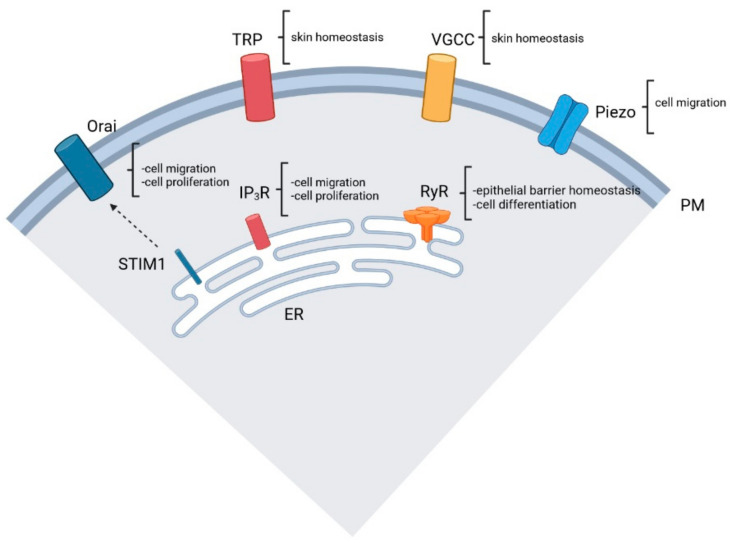
Key type of Ca^2+^ channels. See text for more details. PM: plasma membrane; ER: endoplasmic reticulum. Created with BioRender.com.

## Data Availability

Not applicable.

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
