# Peer review of "Wound Repair and Ca2+ Signalling Interplay: The Role of Ca2+ Channels in Skin"

_cells, 2024, doi:10.3390/cells13060491_

Round 1

Reviewer 1 Report

Comments and Suggestions for Authors

This review describes the role of calcium signaling in wound healing. The main purpose of the manuscript is to discuss the ROLE of various calcium-permeable channels and receptors in regulating the response to injury, migration, phagocytosis, cell proliferation, and extracellular matrix remodeling, which are cellular processes involved in wound healing. However, I have significant concerns about various aspects of the manuscript. Below is a list of issues that should be addressed before the manuscript is published in Cells.

1)   I am not sure why the authors use small paragraphs throughout the text. I got the impression that ideas were not fully developed with this particular format. For example, in section 2, there are 10 paragraphs alone. Similarly, in Pag. 7, there is a one sentence paragraph (lines 288-289). In page 8 there is also a large number of paragraphs containing 1 or 2 sentences. The authors are encouraged to revise the manuscript and have a better layout of the information provided.

2)   I don’t believe that the title corresponds to the content of the review. From the title, we are made to believe that there is a discussion regarding the role of various calcium-permeable ion channels and receptors in the cellular processes, involved in wound healing. However, most of the content just describe the EXPRESSION of various calcium-permeable channels and receptors in the skin (mainly keratinocytes). In very few instances, we see a clear description of how ion channels and receptors REGULATE the response to injury, migration, phagocytosis, cell proliferation, and extracellular matrix remodeling.

3)   There are numerous editorial and grammatical errors throughout the text (proofreading by a native English speaker is recommended). Some of these issues are listed below. However, there are many more editorial and grammatical errors that require a more detailed revision.

a)     Pages 3-4, line 127-135: the list of various TRP channels can be separated by commas as part of a sentence.

b)     Page 3, line 106-107: Unclear sentence “These discoveries prove that VGCCs are related with skin homeostasis and for skin barrier recovery a negative role.”

c)     Page 5, line 197-198: “According to the literature thermo-TRP channels such as TRPV1, 197 TRPV2, TRPV3, and TRPV4 can be triggered by heat”. Please note that channels and receptors are activated or stimulated…they are not triggered!

d)     Page 5, line 199-201: Unclear sentence “On the cells of basal and supra-basal healthy human skin, the TRPV1-4 immunoreactivity has been recognized, and these group has been suggested be thermo-sensory receptors [57,58].”

e)     Page 5, line 212-213: missing comma and “and” in the sentence “Temperature and ligands, such as cannabinoids, lipids and probenecid regulate TRPV2.”

f)      Page 7, line 298-300: Unclear sentence (and extra commas) “Moreover, IP3R engages with Ca2+ in a biphasic way, i.e. activation at low concentrations (up to 0.3 μM) and suppression at higher values (0.5–1 μM) [99] and IP3R function is also controlled by accessory proteins Ca2+ -independent, , redox potential, ATP and Mg2 [100].”

g)     Page 7, line 335-337: Why immune in capital? “Moreover, decreased SOCE in ORAI1-, ORAI2-, and ORAI1/2-deficient Immune cells, for instance neutrophils, affects multiple cellular functions, such as degranulation, leukotriene expression, phagocytosis, and reactive oxygen species (ROS) production [110].”

Comments on the Quality of English Language

The manuscript requires extensive proofreading and editing.

Author Response

This review describes the role of calcium signaling in wound healing. The main purpose of the manuscript is to discuss the ROLE of various calcium-permeable channels and receptors in regulating the response to injury, migration, phagocytosis, cell proliferation, and extracellular matrix remodeling, which are cellular processes involved in wound healing. However, I have significant concerns about various aspects of the manuscript.

We appreciate the reviewer's suggestions. We have made significant changes to the text and updated the English.

Below is a list of issues that should be addressed before the manuscript is published in Cells.

  • I am not sure why the authors use small paragraphs throughout the text. I got the impression that ideas were not fully developed with this particular format. For example, in section 2, there are 10 paragraphs alone. Similarly, in Pag. 7, there is a one sentence paragraph (lines 288-289). In page 8 there is also a large number of paragraphs containing 1 or 2 sentences. The authors are encouraged to revise the manuscript and have a better layout of the information provided.

We have amended the ms and realized a better layout of the text.

  • I don’t believe that the title corresponds to the content of the review. From the title, we are made to believe that there is a discussion regarding the role of various calcium-permeable ion channels and receptors in the cellular processes, involved in wound healing. However, most of the content just describe the EXPRESSION of various calcium-permeable channels and receptors in the skin (mainly keratinocytes). In very few instances, we see a clear description of how ion channels and receptors REGULATE the response to injury, migration, phagocytosis, cell proliferation, and extracellular matrix remodeling.

We have changed the title to reflect the progress of the manuscript more accurately, as advised.

  • There are numerous editorial and grammatical errors throughout the text (proofreading by a native English speaker is recommended). Some of these issues are listed below. However, there are many more editorial and grammatical errors that require a more detailed revision.
  1. Pages 3-4, line 127-135: the list of various TRP channels can be separated by commas as part of a sentence.

We have corrected it

  1. Page 3, line 106-107: Unclear sentence “These discoveries prove that VGCCs are related with skin homeostasis and for skin barrier recovery a negative role.”

We have modified this sentence.

  1. c) Page 5, line 197-198: “According to the literature thermo-TRP channels such as TRPV1, 197 TRPV2, TRPV3, and TRPV4 can be triggered by heat”. Please note that channels and receptors are activated or stimulated…they are not triggered!

We have corrected it

  1. Page 5, line 199-201: Unclear sentence “On the cells of basal and supra-basal healthy human skin, the TRPV1-4 immunoreactivity has been recognized, and these group has been suggested be thermo-sensory receptors [57,58].”

We have modified this sentence.

  1. e) Page 5, line 212-213: missing comma and “and” in the sentence “Temperature and ligands, such as cannabinoids, lipids and probenecid regulate TRPV2.”

We have modified this sentence.

  1. Page 7, line 298-300: Unclear sentence (and extra commas) “Moreover, IP3R engages with Ca2+ in a biphasic way, i.e. activation at low concentrations (up to 0.3 μM) and suppression at higher values (0.5–1 μM) [99] and IP3R function is also controlled by accessory proteins Ca2+ -independent, redox potential, ATP and Mg2 [100].”

We have modified this sentence.

  1. g) Page 7, line 335-337: Why immune in capital? “Moreover, decreased SOCE in ORAI1-, ORAI2-, and ORAI1/2-deficient Immune cells, for instance neutrophils, affects multiple cellular functions, such as degranulation, leukotriene expression, phagocytosis, and reactive oxygen species (ROS) production [110].”

We have modified it

Reviewer 2 Report

Comments and Suggestions for Authors

This review by Bonsignore et al. on the role of Ca2+ signaling on wound repair is certainly of interest, but it is reviewed in an inadequate way since it summarises in a general way the role of calcium channels in wound healing processes without describing it in detail. In this respect a graphical summary displaying the different roles of calcium channels in the wound healing process could be helpful. In summary, this review needs a major revision.

Comments on the Quality of English Language

The syntax of English has to be improved

Author Response

This review by Bonsignore et al. on the role of Ca2+ signaling on wound repair is certainly of interest, but it is reviewed in an inadequate way since it summarizes in a general way the role of calcium channels in wound healing processes without describing it in detail. In this respect a graphical summary displaying the different roles of calcium channels in the wound healing process could be helpful. In summary, this review needs a major revision.

We appreciate the reviewer's suggestions. Two figures have been added, and the text of the ms has been significantly modified.

Reviewer 3 Report

Comments and Suggestions for Authors

This is a comprehensive and informative review. It reads well and is easy to follow.

I do think there is a lot of text and it might be helpful for the reader to have a figure showing the various calcium channels discussed, and hope they interact (e.g. Orai1 and TRPV6). 

One minor point: line 344; should be rapid not solid

Comments on the Quality of English Language

Minor editing in places

Author Response

This is a comprehensive and informative review. It reads well and is easy to follow.

I do think there is a lot of text and it might be helpful for the reader to have a figure showing the various calcium channels discussed, and hope they interact (e.g. Orai1 and TRPV6).

One minor point: line 344; should be rapid not solid

We appreciate the reviewer's favorable assessment of our work. Two figures have been added, and the text has been modified.

Round 2

Reviewer 1 Report

Comments and Suggestions for Authors

I appreciate the revised version of the manuscript with added illustrations. The authors made significant improvements in the revised version of the manuscript, but some issues still persist. There are still numerous editorial and grammatical errors throughout the text (proofreading by a native English speaker is recommended). Some of these issues are listed below. However, there are many more editorial and grammatical errors that require a more detailed revision.

a)     Pages 4, lines 202-206: “Depending on their structure, TRP channels can be classified into different subtypes. There are seven subfamilies: TRPA (ankyrin), TRPC (canonical), TRPM (melastatin), TRPML (mucolipin), TRPN (Drosophila NompC), TRPP (polycystin), and TRPV (vanilloid), [28,37,38].”

b)     Pages 4, lines 230-221: “An increase in Ca2+ entry provokes the depolarization of the  membrane and an increase in cytosolic Ca2+ concentration”

c)     Page 9, lines 2725-2730: Unclear summary “In summary, some works have displayed the potential of incorporating and controlling Ca2+ release from scaffolds to accelerate wound healing [124]. But, more preclinical and clinical data are mandatory to validate the findings of studies concerning the ability to modulate Ca2+ signaling in the wound area to obtain a better repair and tissue restoration.”

Comments on the Quality of English Language

Please see my comments above.

Author Response

I appreciate the revised version of the manuscript with added illustrations. The authors made significant improvements in the revised version of the manuscript, but some issues still persist. There are still numerous editorial and grammatical errors throughout the text (proofreading by a native English speaker is recommended). Some of these issues are listed below. However, there are many more editorial and grammatical errors that require a more detailed revision.

We appreciate the reviewer's good assessment of our manuscript. We have currently improved the English quality.

  1. Pages 4, lines 202-206: “Depending on their structure, TRP channels can be classified into different subtypes. There are seven subfamilies: TRPA (ankyrin), TRPC (canonical), TRPM (melastatin), TRPML (mucolipin), TRPN (Drosophila NompC), TRPP (polycystin), and TRPV (vanilloid), [28,37,38].”

We have corrected it.

  1. b) Pages 4, lines 230-221: “An increase in Ca2+ entry provokes the depolarization of the membrane and an increase in cytosolic Ca2+ concentration”

We have modified this statement.

  1. Page 9, lines 2725-2730: Unclear summary “In summary, some works have displayed the potential of incorporating and controlling Ca2+ release from scaffolds to accelerate wound healing [124]. But, more preclinical and clinical data are mandatory to validate the findings of studies concerning the ability to modulate Ca2+ signaling in the wound area to obtain a better repair and tissue restoration.”

We have modified this statement.

Reviewer 2 Report

Comments and Suggestions for Authors

The authors followed the advices by the reviewer, the revised manuscript could be published after careful editing of the English.

Comments on the Quality of English Language

Quality of English should be improved

Author Response

The authors followed the advices by the reviewer, the revised manuscript could be published after careful editing of the English.

We appreciate the reviewer's favorable assessment of our manuscript. We've now raised the text's quality.